# Integrating Remote Sensing and Street View Images to Quantify Urban Forest Ecosystem Services

**Elena Barbierato [1], Iacopo Bernetti [1,\*] , Irene Capecchi [1] and Claudio Saragosa [2]**

1   Department of Agriculture, Food, Environment and Forestry University of Florence, Piazzale delle Cascine 18, 50144 Firenze, Italy; elena.barbierato@unifi.it (E.B.); irene.capecchi@unifi.it (I.C.)
2   Department of Architecture, University of Florence, Via della Mattonaia, 14, 50121 Firenze, Italy; claudio.saragosa@unifi.it
\*   Correspondence: iacopo.bernetti@unifi.it; Tel.: +39-3298603416

**Abstract:** There is an urgent need for holistic tools to assess the health impacts of climate change mitigation and adaptation policies relating to increasing public green spaces. Urban vegetation provides numerous ecosystem services on a local scale and is therefore a potential adaptation strategy that can be used in an era of global warming to offset the increasing impacts of human activity on urban environments. In this study, we propose a set of urban green ecological metrics that can be used to evaluate urban green ecosystem services. The metrics were derived from two complementary surveys: a traditional remote sensing survey of multispectral images and Laser Imaging Detection and Ranging (LiDAR) data, and a survey using proximate sensing through images made available by the Google Street View database. In accordance with previous studies, two classes of metrics were calculated: greenery at lower and higher elevations than building facades. In the last phase of the work, the metrics were applied to city blocks, and a spatially constrained clustering methodology was employed. Homogeneous areas were identified in relation to the urban greenery characteristics. The proposed methodology represents the development of a geographic information system that can be used by public administrators and urban green designers to create and maintain urban public forests.

**Keywords:** urban forest; landscape metrics; LiDAR; aerial images; street view images; semantic segmentation; convolutional neural network (CNN); spatial clustering

## 1. Introduction

Holistic tools to assess the health impacts from climate change mitigation/adaptation policies enacted to increase the amount of public green spaces are urgently needed. Urban vegetation provides numerous ecosystem services on a local scale and is therefore a potential adaptation strategy that can be monopolized in the era of global warming, and it can also offset the increasing impacts of human activity on the urban environment.

In this respect, on their review on health and climate relating to ecosystem services provided by street trees in an urban environment, Salmon et al. [1] stated that, "Our review, in agreement with other papers in the ecosystem services (ESS) literature...has also highlighted the importance of scale when determining the effect of trees on climate and health. Whilst much of the research to date has focused on the regional and urban scale effects of vegetation on climate and health, it is much less clear what the impacts of street trees are at local scales where the result of the intervention is most clearly felt".

On a local scale, the presence of street trees can modify indoor temperatures by shading buildings and can significantly reduce the risk of indoor overheating [2]. An empirical study [3] conducted on a project scale using direct measurements obtained with an infrared camera showed that the degree of

foliage shading from rows of trees on building facades may decrease surface temperatures by up to 9 °C.

Streiling and Matzarakis [4] found that clustering trees into lines or small groups interspersed with open areas can help reduce the radiative load, provide shade, and allow long-wave cooling at night. Growing large and broad trees with dense canopies can be considered in streets that have a low height/width ratio, while taller narrower trees can be grown in streets that have a high height/width ratio [5]. On a local scale, the characteristics of tree canopy, tree density, and proximity to other urban structures influence the ability of plants to remove pollutants [6,7] Nilsson et al. [8] showed that in the ecosystem service of traffic noise attenuation, the urban green characteristics on a local scale (depth, width, and stem diameter of a tree belt) play a fundamental role. In addition, with respect to cultural ecosystem services, a laboratory psychometric study was conducted in which three-dimensional (3D) videos produced in a laboratory contained urban green coverage at eye level ranging from 0% to 70%: videos that contained a higher level of public greenery elicited a greater self-reported stress reduction [9].

To transpose the results of previous research to other cities, specific ecological metrics on a local scale are needed, and a combination of traditional remote sensing and so-called "proximate sensing" appears to be a good candidate in this respect. The traditional field of remote sensing uses overhead images of distant scenes to derive geographic information, and proximate sensing uses ground-level images of objects and scenes that are nearby [10].

When classifying land cover, urban tree cover has traditionally been quantified using long-range, remotely-sensed image processing, such as satellite imagery (LANDSAT), ortho-aerial photographs or, more recently, Laser Imaging Detection and Ranging (LiDAR) [11,12], or by employing data derived from field surveys [13]. Therefore, although the ecological metrics calculated from remote images can be useful to quantify some ecosystem services, they are not applicable for assessing ecosystem services that depend on street-level urban greening measures. Earth observation data, such as multispectral satellite images, have long been used to classify green open spaces in cities. Despite the large increase in spatial resolution, urban green classification from aerial images is still considered a difficult task, since only the upper part of plants can be captured from the nadir's point of view. For this reason, high resolution satellite images are useful for classifying urban green with a wide spatial extension, such as urban forests, green parks and gardens, but they are not efficient for urban green scattered or in rows. In fact, the lack of detail at ground level makes it difficult to detect the characteristics deriving from the shape of the foliage of the plants. Therefore, the urban green classification at street level is still based on a labor-intensive territorial survey, which is inefficient and expensive. Fortunately, the growing accessibility to different geo-tagged data sources allows to merge remote sensing images with data of different modalities and observations. For example, Google Street View (GSV) offers users street-level panoramic images captured in thousands of cities around the world, which allows you to observe street scenes in big cities and thus provides instant detection capabilities and detail at the level of the soil that lack in aerial imagery. Recently, Li et al. [14,15] developed a novel Google Street View (GSV)-based method to study the distribution of street greenery. Unlike green metrics derived from remotely-sensed data, the GSV-based method quantifies street greenery using a perspective from the ground, which better represents the distribution of street greenery projected on building facades or on street pavements. The aims of previous work have been to analyze the relationship between perceived safety and green vegetation characteristics [14,16], quantify the sky view factor (SVF) from street-level imagery, and to assess environmental inequalities in terms of different types of urban greenery [17].

The aim of this current study is to create a general-purpose set of ecological metrics by combining remote sensing and proximate sensing (Street View) approaches with data retrieved from GSV, to quantify urban forest ecosystem services and provide a widely transferable methodology. In this respect, remote sensing metrics were calculated by combining high-resolution multispectral images and LiDAR data to produce indices at different altitudes with respect to the ground. The ecological metrics from proximate sensing were then calculated by semantic segmentation using pretrained deep

neural networks. To estimate the validity of this approach, a set of ecological metrics was used to classify contiguous homogeneous areas of a city through a spatial clustering algorithm [18,19]. Figure 1 presents a diagram of the workflow.

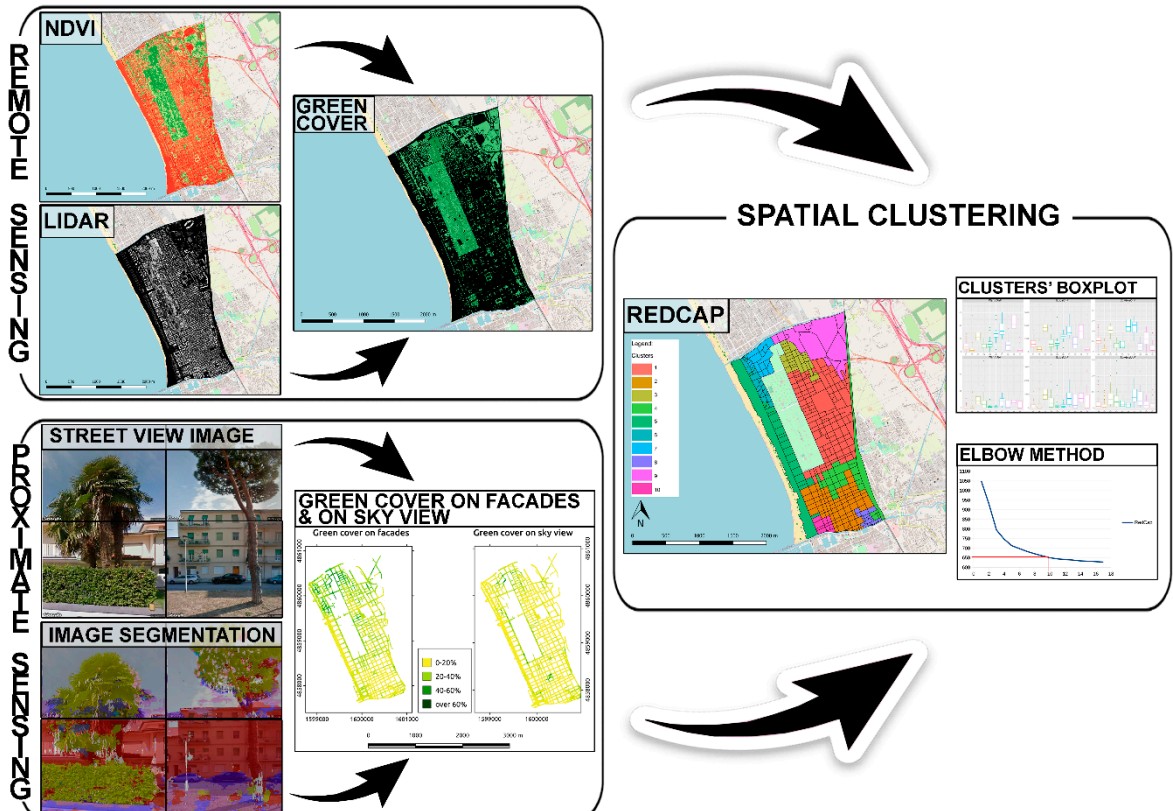

**Figure 1.** Workflow diagram.

## 2. Materials and Methods

### 2.1. Study Area and Data Sources

#### 2.1.1. Study Area

Our research was conducted in the city of Viareggio in northern Tuscany, Italy. The study area has boundary coordinates (datum WGS84, projection Universal Transverse of Mercator (UTM), zone 32) Nord min = 598,681, Nord max = 601,019, East min = 4,857,573, East max = 4,860,971, and mean latitude = 43.87911° N.

Viareggio has a population of over 62,000 and is a seaside tourist town on the coast of the Ligurian Sea. It is characterized by orthogonal streets that form rectangular blocks, and the building types are terraced houses (with one or two floors), villas, and hotels. Urban greenery is widespread throughout the city, but it has different typologies. In the northern area, there is a greater percentage of greenery (hedges and rows of large trees higher than the facades of buildings), and in the southern area, small trees are spaced further apart and are at a lower height than the facades of buildings (Figure 2). The perimeter of the study area is defined by both natural and artificial borders: in the north, east, south, and west are the Fosso dell 'Abate waterway, the railway, the Burlamacco canal, and the Ligurian Sea, respectively.

The study area covers an area of 3,555,104 square meters with 768,434 square meters of urban green and the remaining part by artificial surfaces.

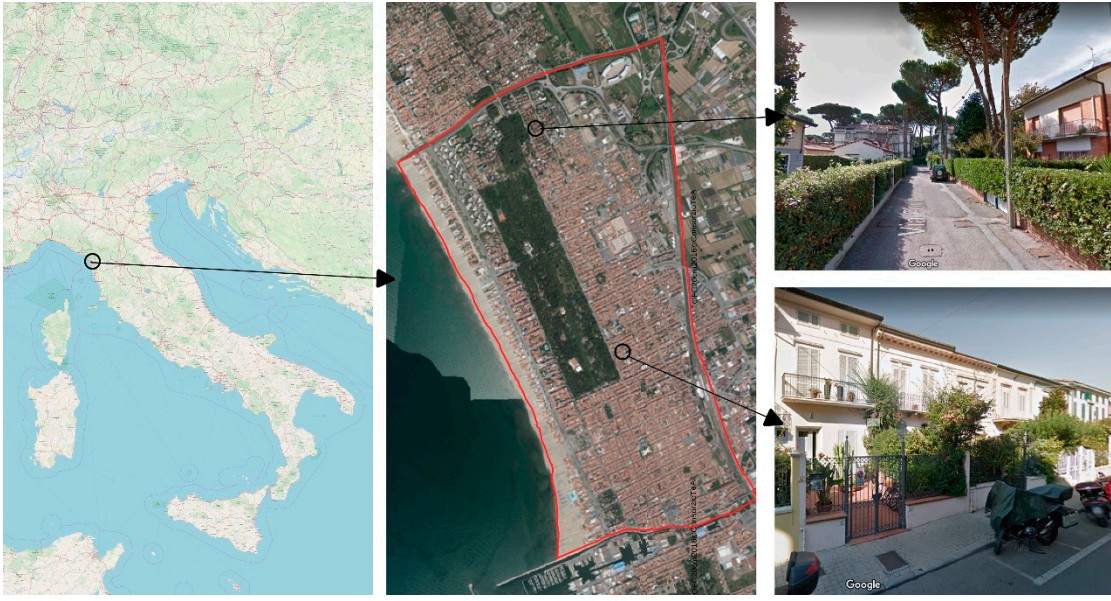

**Figure 2.** Study Area.

### 2.1.2. Data Sources

The input data were both cartographic information and remote sensing data. Remote sensing data were downloaded from a database of the Tuscan region, and vegetation cover data were obtained from photogrammetric processing of seven aerial multispectral frames (four bands: red, green and blue + near infrared regions (NIR) acquired on October 2013 using an UltraCam Xp (Vexcel) digital metric camera (resolution of 0.2 × 0.2 m). UltraCam Xp simultaneously collects light from five different spectral bands. The spectral sensitivity of red, green, blue, and near infrared and the panchromatic channel from 410 nm to 690 nm, RED from 580 nm to 700 nm, GREEN from 480 nm to 630 nm, BLUE from 410 nm to 570 nm, and NIR from 690 nm to 1000 nm.

The heights of vegetation and buildings were derived from 9 LiDAR image on 2006 (resolution of 1 × 1 m). The LiDAR data were provided by the Italian Ministry of the Environment, Land, and Sea. The points acquired from this survey have an altimetric accuracy of ±15 cm and a planimetric accuracy is ±30 cm. In this work, the data available by the geographical portal of the Tuscan region with a resolution of 1 × 1 m were used. That resolution was considered satisfactory for the objectives of the work. If necessary, however, the proposed method could therefore also be applied to more detailed scales.

The other cartographic data were derived from a topographic regional database in 2013 with detail on a scale of 1:2000. All the input raster (multispectral images and city blocks raster) were aligned at a 1 × 1 m resolution using the nearest neighbor algorithm.

### 2.1.3. The City Block

In this work, the reference cartographic element is the city block, as it is the central element of urban planning and urban design, and the basic building block of an urban city. We analyzed vegetation within city blocks. According to the Oxford Dictionary definition, a city block is the smallest area that is surrounded by four streets, usually containing several buildings and vegetation, and urban block boundaries are frequently used to define units for extracting metrics from remotely-sensed data [20]. In according to the definition of Oxford Dictionary, a city block is the smallest area that is surrounded by four streets, usually containing several buildings and vegetation. The blocks were obtained by clipping the study area using OpenStreetMap roads.

### 2.2. Proximate Sensing Landscape Metrics

#### 2.2.1. Google Street View (GSV) Images Collection

Leung and Newsam use the term "Proximate Sensing" to describe a more comprehensive framework that uses ground level images of nearby objects and scenes to automatically map what-is-where on the surface of the Earth, similar to how remote sensing uses overhead images [21]. In this study, we used GSV images downloaded using the GSV static imageApplication programming interface API.

By specifying different parameters in the API, users can download GSV images with different fields of view, heading angles, and pitch angles. In this respect, heading indicates the compass heading of the camera, (heading values range from 0 to 360), pitch specifies the up or down angle of the camera relative to the street view vehicle, and the field of view determines the horizontal field view of the image. These parameters were used to define the ecological metrics of the streetscape.

According to literature, the ecological services of urban green areas are linked to two effects: the shading of foliage on the facades of buildings and the coverage of the sky view of the street [1,3–5,7]. For this reason, we defined two ecological metrics as follows: the percentage of green cover on facades (GCF), which is defined as Equation (1):

$$GCF = \left( \frac{\sum number\ of\ pixels\ classified\ a\ street\ inside\ FOV_{facades}}{\sum total\ number\ of\ pixels\ inside\ FOV_{facades}} \right), \tag{1}$$

And percentage of green cover on sky view of the street (GCS), which is defined as Equation (2):

$$GCS = \left( \frac{\sum number\ of\ pixels\ classified\ a\ street\ inside\ FOV_{sky}}{\sum total\ number\ of\ pixels\ inside\ FOV_{sky}} \right). \tag{2}$$

where $FOV_{facades}$ and $FOV_{sky}$ are the field of view of the image of the facades of buildings and the field of view of the sky view of the street respectively, and $l$ and $r$ are the left and right sides of the street.

Images from all image collection points along city roads were downloaded every 15 m along each roadway, although there were no data for some GSV sampling points, road segments, and areas of the city for various reasons (such as corrupt or missing data or areas with no-coverage). Notwithstanding those instances, the sampling regime covered the full extent of the cities' official boundaries. Four ecological metrics can be extracted for each GSV sampling point: two for the right side and two for the left side.

The following methodology was used to define the parameters necessary for extracting the four GSV images: first, we linked the distances of headings and heights relative to the buildings on the right and left with the GSV sampling points. The buildings' heights were calculated using a map overlay operation between LiDAR geodata and the building layer of the Open Street Map geodatabase. The procedure used to link the geometric parameters of the streetscape (headings, heights, and distances) with the GSV sampling points (Figure 3a) was conducted using Geographic Resources Analysis Support System GRASS software and is available as supplementary material (file grassProcedure.txt). The $FOV_{facades}$ and the $FOV_{sky}$ were then calculated using the following formula (Figure 3b), respectively:

$$FOV_{facade}^{l,r} = 2 \cdot \tan^{-1} \left( \frac{h_{facades}^{l,r} - h_{google}}{l_{street}^{l,r}} \right) \tag{3}$$
$$FOV_{sky}^{l,r} = 90 - FOV_{facade}^{l,r}$$

Finally, the pitch angles relative to the two metrics were obtained using the following:

$$pitch_{facade}^{l,r} = 0 \tag{4}$$
$$pitch_{sky}^{l,r} = \frac{FOV_{facade}^{l,r}}{2} + \frac{FOV_{sky}^{l,r}}{2} \cdot$$

Each digital photograph (red-green-blue color channel jpeg image) was acquired from the GSV API at a resolution of 400 by 400 pixels. The images were downloaded via the "Googleway" library using the statistical software, R. The procedure used is available as supplementary material (file R_procedure.R).

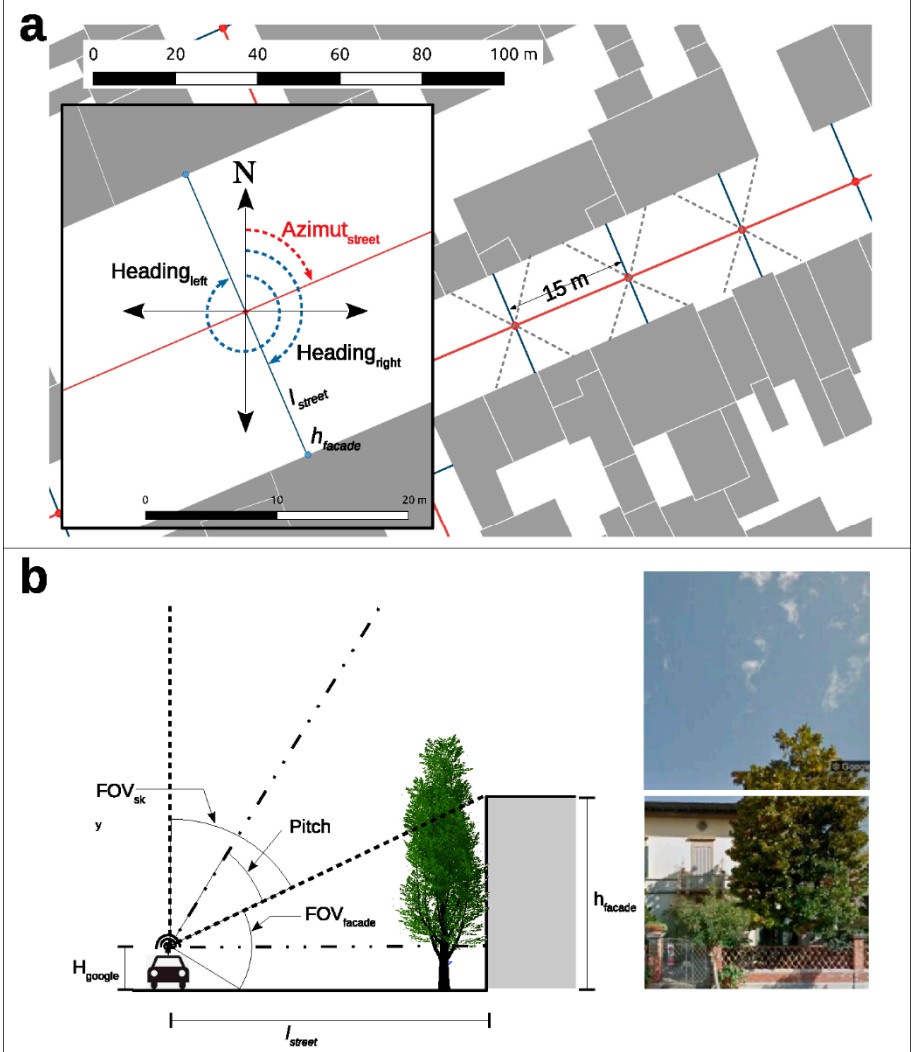

**Figure 3.** Sampling procedure: (**a**) sampling points, (**b**) Pitch and Field of View (FOV).

2.2.2. Image Segmentation

We estimated the total area covered by trees in each image by applying semantic segmentation using deep learning [22]. A semantic segmentation network classifies every pixel in an image, which results in an image segmented by class. In this phase of the work, we used the pre-trained network of Matrix Laboratory MATLAB software based on the Deeplabv3+ network, which is one type of convolutional neural network (CNN) designed for semantic image segmentation [23], with weights initialized by a pre-trained ResNet-18 network. ResNet-18 is an efficient network that is well suited to applications that have limited computing resources. The network was trained using the CamVid dataset [24] from the University of Cambridge, which is a collection of images containing street-level views obtained while driving, and it provides pixel-level labels for 32 semantic classes including car, pedestrian, and road. To make the training easier, the 32 original classes in CamVid were grouped into 11 classes as follows: "Sky", "Building", "Pole", "Road", "Pavement", "Tree", "SignSymbol", "Fence", "Car", "Pedestrian", and "Cyclist".

To validate the network, we extracted 200 random images from those downloaded with Street View API, manually segmented them, and then used them as a validation set to evaluate the performances of the pretrained network. Deeplab v3+ is trained using 60% of the images from the dataset. The rest of the images are split evenly in 20% and 20% for validation and testing, respectively.

### 2.3. Remote Sensing Landscape Metrics

The remote sensing data were used to obtain the coverage and height of vegetation. The urban vegetation coverage was identified through an analysis of the normalized difference vegetation index (NDVI). As reported in the literature [25,26], since only healthy vegetation was included in the study, it was extracted with respect to the NDVI having a value greater than or equal to 0.2. The result was presented as a Boolean map with a resolution of 1 m (which is similar to LiDAR data), in which the value of 0 indicates an absence of vegetation, while a value of 1 indicates the presence of vegetation.

Since the urban green area can be characterized by various types of vegetation with differing heights, shapes, and ecological functions, we distinguished two types according to the average height value of the buildings in each city block. To obtain the height of the vegetation, we overlaid the NDVI binary and normalized digital surface model (nDSM) generated from LIDAR data, which provided a raster map divided into two height classes (Figure 4): the first class is green cover on facades, and is represented by a value less than (or equal to) the average height of the buildings, and the second is green cover seen on a sky view, and is represented by the average value of the height of the buildings.

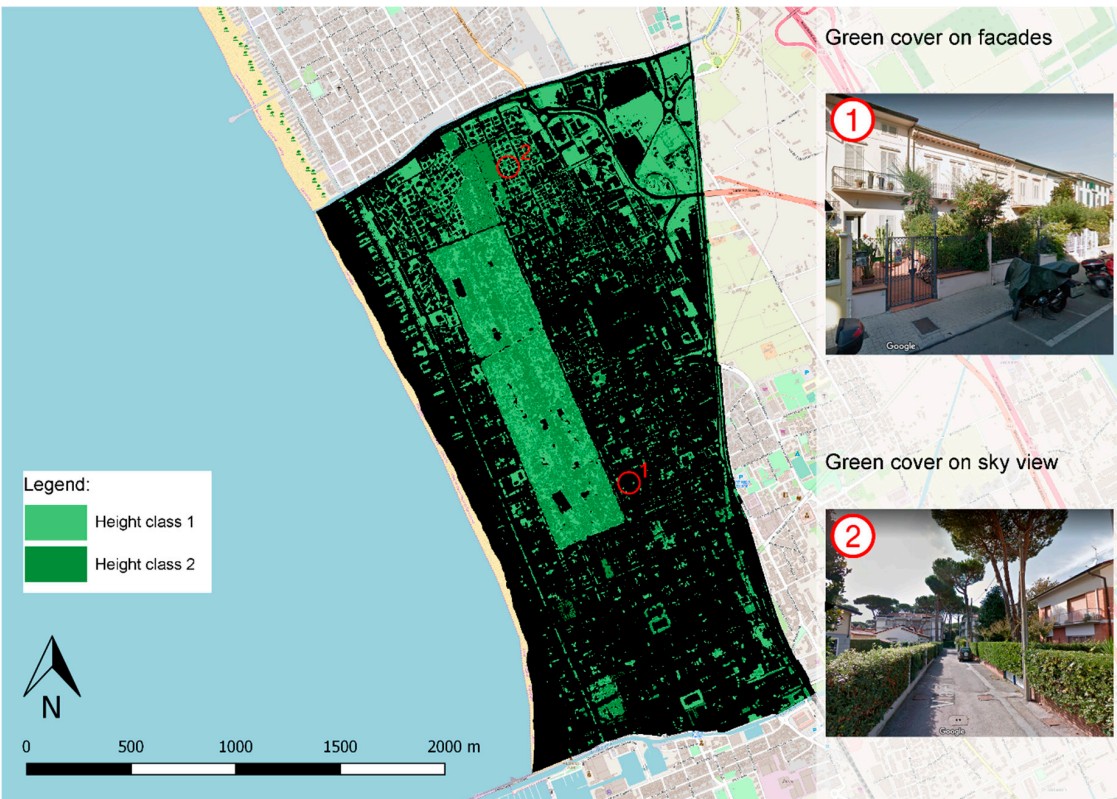

**Figure 4.** Urban green map classification.

The results were spatialized on the rasterized city blocks, as they are the central elements of urban planning and urban design on which the landscape metrics were calculated. As we considered that the ecological characteristics of urban greenery depend not only on the overall surface coverage, but also on the shape and distribution of vegetation within city blocks, we identified homogeneous city blocks through the use of landscape metrics. The metrics we used were the percentage of landscape (PLAND) and edge density (ED), which were normalized for the area. All metrics was calculated using

Fragstats 4.2 software. We chose PLAND because the city blocks had different dimensions, while ED is an important ecological parameter and many urban green benefits (such as pollution abatement, acting as an acoustic barrier, and being aesthetically pleasing) depend on the linear development and distribution of vegetation rather than its surface attributes [27–29].

Similar to the metrics derived from proximate sensing, the urban green metrics obtained from remote sensing were also calculated by classifying urban greenery into two classes through a map overlay operation between LiDAR and NDVI data. The two classes are: below the height of the block buildings and above the height of block buildings.

PLAND enables the percentage plant cover in each city block to be determined using the following calculation:

$$PLAND_i = \frac{\sum_{j=1}^{n} a_{ij}}{A} \tag{5}$$

where $PLAND_i$ is the proportion of the landscape occupied by a patch type (class), $i$ (below or over the height of buildings), $a_{ij}$ is the area of patch $i$, $j$, and $A$ is the total landscape area.

The $ED_i$ metric enabled the compactness and distribution of the vegetation of each block to be determined using the following calculation:

$$ED_i = \sum_{k=1}^{m} e_{ik} \tag{6}$$

where $e_{ik}$ is the total length (m) of the landscape edge involving patch type (class), $i$, and includes the landscape boundary and background segments involving patch type $i$.

Spatial Clustering

As the ecological and visual characteristics of a city are manifested on a larger scale than that of a city block, it was necessary to create homogeneous areas by clustering city blocks based on their urban greenery characteristics.

As traditional clustering procedures do not consider the spatial relation between the geometries [30], we used a spatially bound geographic clustering procedure that grouped the territorial area objects into homogeneous contiguous regions [31].

Regionalization with dynamically constrained agglomerative clustering and partitioning (REDCAP) is a new method of spatial clustering and regionalization that was presented by Guo [32]. It is essentially based on a group of six methods for regionalization that are composed using a combination of three agglomerative clustering methods (single linkage clustering, SLK, average linkage clustering, AVG, and complete linkage clustering, CLK) and two different spatial constraining strategies: first-order constraining and full-order constraining [32]. The work of Guo [32] describes the technical and computational details of these six methods of regionalization, but we briefly describe here the theoretical context in which REDCAP is collocated, how it works in the case, and how it is applied in our analysis, of the AVG full-order method. The analysis is based on a contiguity matrix and a set of constrained strategies that drive the agglomerative clustering method. The average linkage clustering (ALK) defines the distance between two clusters as the average dissimilarity between all cross-cluster pairs of data points:

$$d_{ALK}(L, M) = \frac{1}{|L||M|} \sum_{u \in L} \sum_{u \in M} d_{uv} \tag{7}$$

where $|L|$ and $|M|$ are the number of data points in clusters L and M, respectively, $u \in L$ and $v \in M$ are two data points, and $d_{uv}$ is the dissimilarity between $u$ and $v$.

The merging process incorporates the contiguity constraints using the full-order constraining strategy [32]. Contiguity-constrained agglomerative clustering requires that two clusters cannot be merged if they are not spatially contiguous, and this is the differential element between classic spatial clustering and regionalization. A full-order constraining strategy includes all edges in the

clustering process, and the distance between two clusters is defined over all edges. This strategy is dynamic because it updates the contiguity matrix after each merge to track all edges which connect two different clusters.

## 3. Results

### 3.1. Image Segmentation

Using the Googleway library, we downloaded 14,542 geo-tagged images relating to 3638 sampling points acquired in 2018. Figure 5 shows the sampling points (Figure 5a) and some typical examples of the segmentation process (Figure 5b). The results shown in Table 1 show an overall accuracy of 89% and an accuracy of 83% for the tree classes.

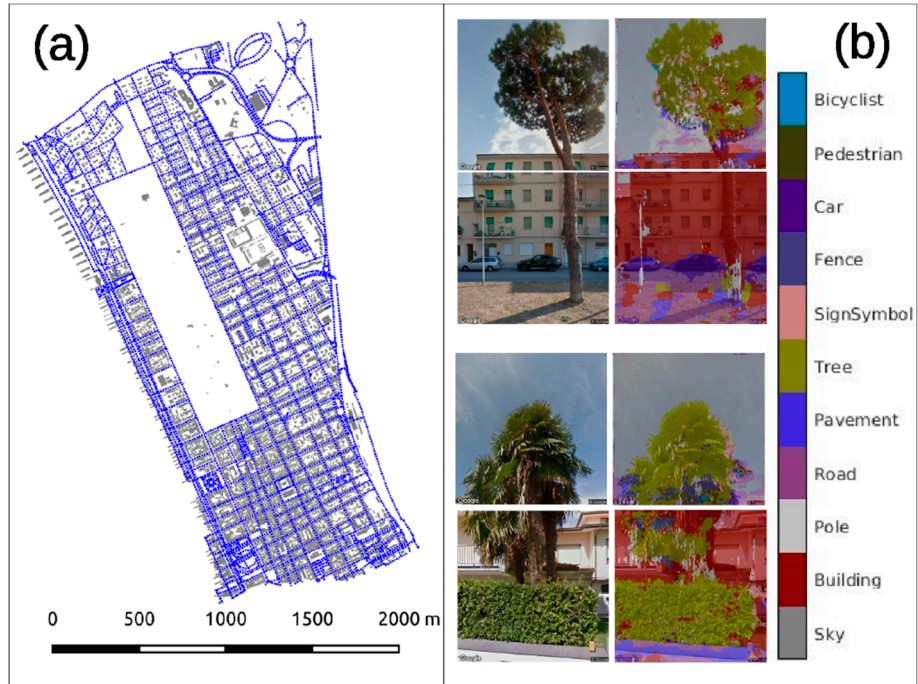

**Figure 5.** (**a**) Sampling points and (**b**) segmentation samples.

**Table 1.** Confusion matrix for network validation.

|  | Sky | Building | Pole | Road | Pavemnt | Tree | Sign. | Fence | Car | Pedestrian | Bicyclist. |
|---|---|---|---|---|---|---|---|---|---|---|---|
| Sky | 0.94 | 0.01 | 0.02 | 0.00 | 0.00 | 0.04 | 0.00 | 0.00 | 0.00 | 0.00 | 0.00 |
| Building | 0.01 | 0.80 | 0.07 | 0.00 | 0.01 | 0.02 | 0.05 | 0.02 | 0.01 | 0.01 | 0.00 |
| Pole | 0.01 | 0.07 | 0.77 | 0.00 | 0.01 | 0.03 | 0.05 | 0.03 | 0.01 | 0.02 | 0.00 |
| Road | 0.00 | 0.00 | 0.00 | 0.94 | 0.04 | 0.00 | 0.00 | 0.00 | 0.01 | 0.00 | 0.00 |
| Pavement | 0.00 | 0.01 | 0.01 | 0.02 | 0.93 | 0.00 | 0.00 | 0.01 | 0.01 | 0.01 | 0.00 |
| Tree | 0.02 | 0.02 | 0.03 | 0.00 | 0.00 | 0.88 | 0.01 | 0.03 | 0.00 | 0.00 | 0.00 |
| SignSymbol | 0.00 | 0.04 | 0.05 | 0.00 | 0.00 | 0.01 | 0.88 | 0.01 | 0.01 | 0.00 | 0.00 |
| Fence | 0.00 | 0.01 | 0.02 | 0.00 | 0.00 | 0.01 | 0.00 | 0.93 | 0.01 | 0.01 | 0.00 |
| Car | 0.00 | 0.00 | 0.01 | 0.00 | 0.00 | 0.00 | 0.01 | 0.02 | 0.91 | 0.03 | 0.01 |
| Pedestrian | 0.00 | 0.02 | 0.02 | 0.00 | 0.01 | 0.00 | 0.00 | 0.01 | 0.02 | 0.90 | 0.01 |
| Bicyclist | 0.00 | 0.00 | 0.01 | 0.00 | 0.01 | 0.00 | 0.00 | 0.00 | 0.01 | 0.02 | 0.95 |

Figure 6 shows the receiver operation curves (ROC) curves and the values of the area under the ROC curve (AUC) for the different classes calculated by means of a sub-sample of 50 images. The area under the curve for all the four class is very large indicating the high accuracy of the algorithm.

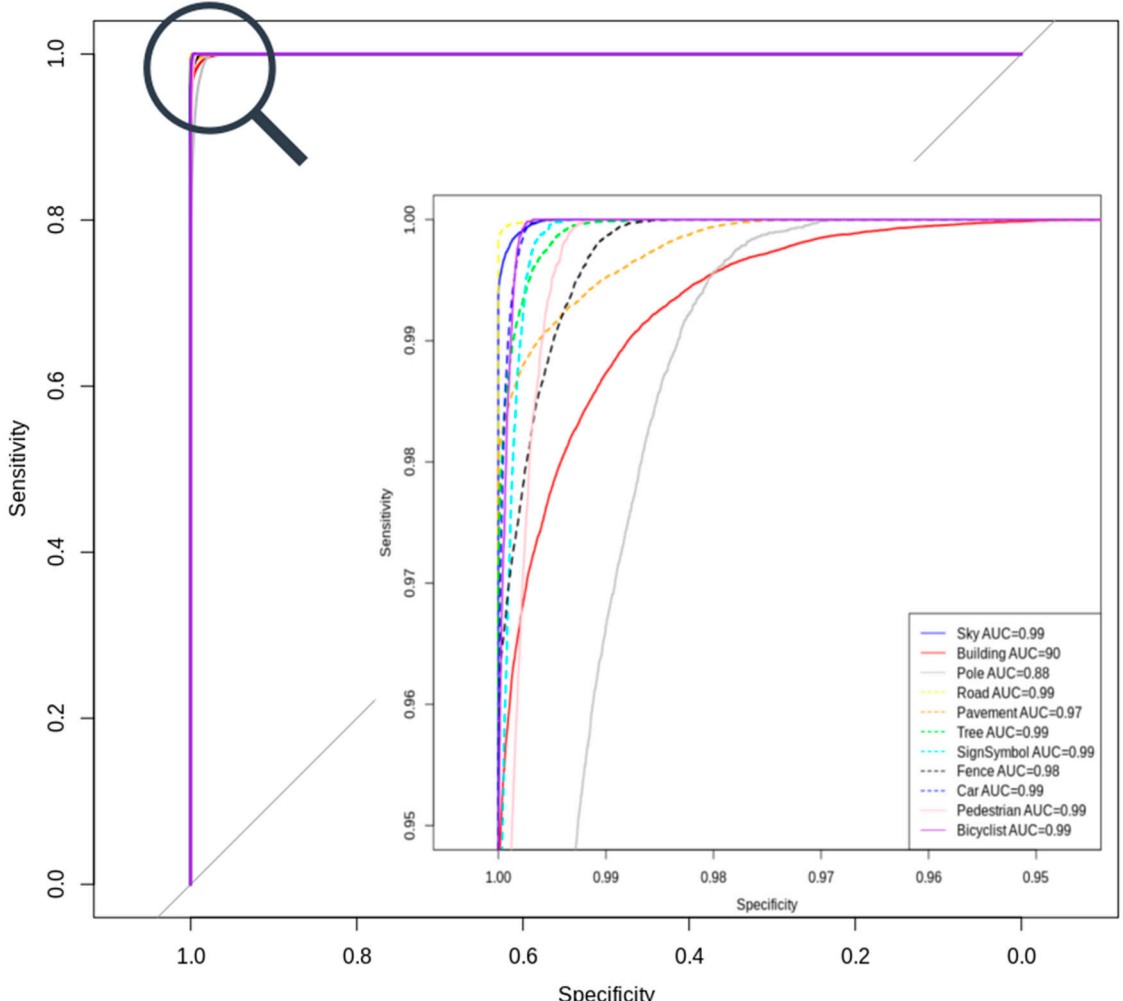

**Figure 6.** Left: receiver operation characteristics (ROC) curves obtained for DeepLabV3+ architecture for semantic segmentation. Zoomed view of the selected area is shows in the right figure. Area under the curve (AUC) values for each method is reported in the legend.

These results are in line with those obtained in other researches that have used DeepLabV3+ in remote sensing [33] or in other fields of study [34]. Few segmentation outputs obtained using DeepLabV3+ are in the supplementary materials. It can be seen that the green class is segmented accurately with a sharp class boundary.

The Figure 7a shows the frequency distribution of the two metrics derived from segmentation of street view images. The city of Viareggio is characterized by greenery affecting most of the lower facades of buildings on public roads. The GCF index has an average value of 10% (median = 8.6%) with the first quartile of 2.9% and the third quartile of 14.6%. However, the GCS index has lower values, with an average of 3.1% (median 1.3%) and first and third quartiles of 0.2% and 4.3%, respectively.

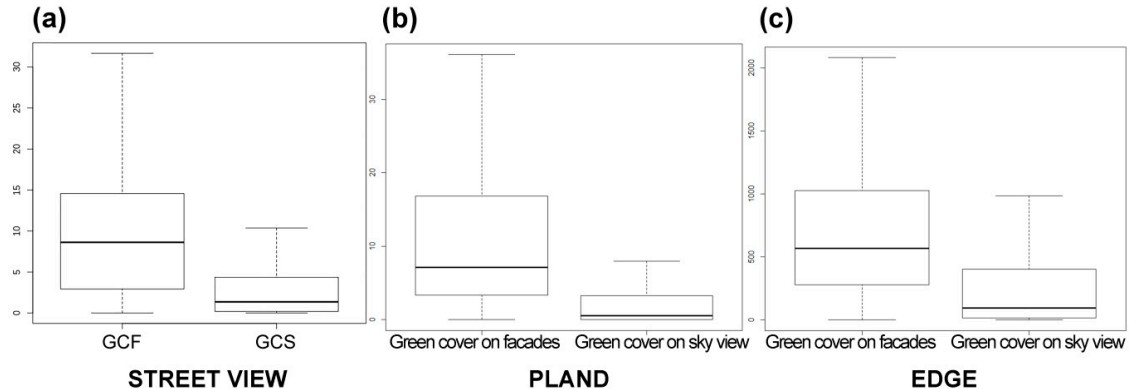

**Figure 7.** Frequency distribution metrics: (**a**) street view metrics (**b**) percentage of landscape (PLAND) and (**c**) edge density (EDGE).

We used the network-based inverse distance weighting (NT-IDW) [35] method to spatialize the relative sampling point values in raster maps of the two indices. The NT-IDW method expands the commonly used spatial interpolation methods, IDW (inverse distance weighting) and inverse distance weighting, and the results are applied to analyze spatial data observed on a network. IDW assumes that all locations exist in a two-dimensional Euclidean space, (the distance between the sample locations and the target locations are measured using the straight-line distance). NT-IDW extends from IDW but uses a network distance instead. NT-IDW was conducted using the routines in the ipdw R package [36], and Figure 8 shows the results of spatialization using the NT-IDW method. The GCF index has higher values in the northern part of the city, especially in the north-west quadrant, where the so-called "garden city" is located. In contrast, the GCS index has relatively high values within isolated hot spots scattered across the entire urban area.

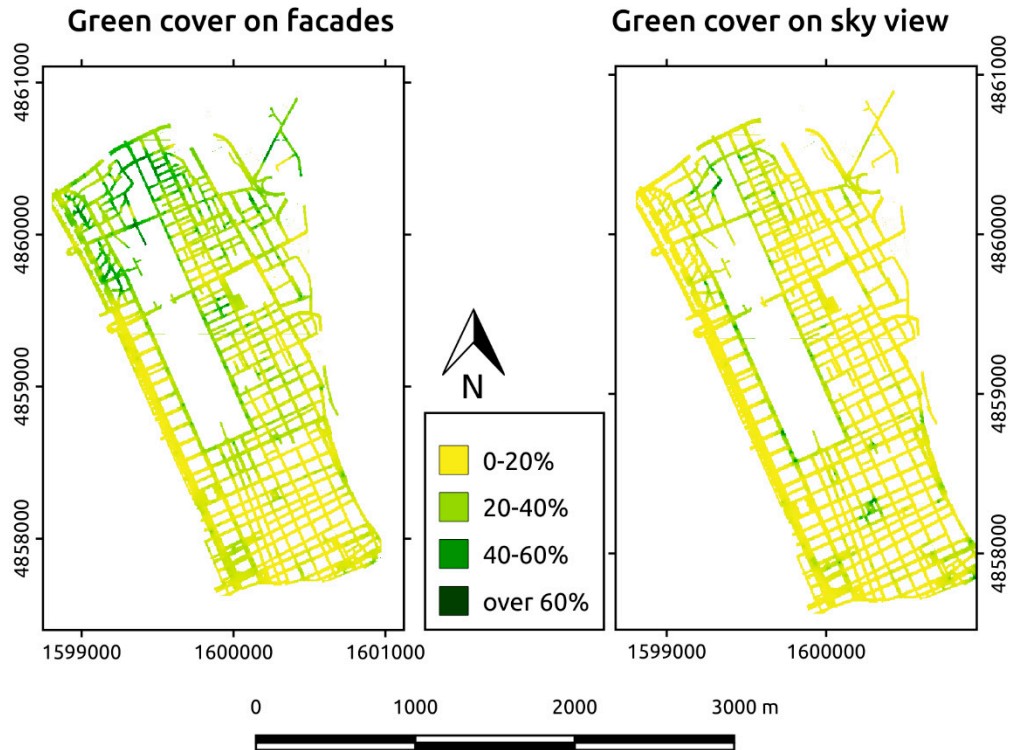

**Figure 8.** Maps of Grenn Cover on Facades (GCF) and Green Cover on Sky (GCS) indices.

### 3.2. Remote Sensing Landscape Metrics

The Figures 7b,c show box and whisker plots of landscape metrics distribution of the green cover on facades compared to the green cover on sky view for PLAND and ED, respectively.

The results of PLAND (Figure 7b) show the percentage of green cover and also show that most of the city has green cover below the height of the buildings rather than higher than buildings. The former has a first quartile value of 3%, median value of 7%, and third quartile value of 17%, while the latter has a first quartile of 0%, median value of 0.5%, and third quartile value of 3%.

The results of ED (Figure 7c) show the no-compactness values of vegetation, which are negatively related to the compactness of urban green. An analysis of the boxplot shows that the city is mostly has green cover below the height of buildings (first quartile value of 280 m/ha, median value of 567 m/ha, third quartile value of 1028 m/ha) but less green cover over the height of buildings (first quartile value of 16 m/ha, median value of 94.5 m/ha, and third quartile value of 405 m/ha).

These results are visually represented through two maps that show the quantile distribution of different vegetation types: green cover on facades and on sky view (Figure 9). In both, the green cover below the height of buildings is predominantly located north of the study area, and its distribution is more compact than that of sky view, which is mainly located south of the city and is more fragmented.

In order to evaluate, the homogeneity of the metrics within each block were calculated by the standard deviation statistics within and between the blocks. In fact, the verification of the homogeneity of the metrics within the blocks allows to verify the efficiency of the blocks as a tessellation element of the study area. The results shown in Table 2 show that the blocks are an efficient tessellation of the urban space. The descriptive statistics of the 351 isolates are available as supplementary material.

**Table 2.** Statistics of metrics within and between the blocks.

| Variable | Within block Mean Square (a) | Between block Mean Square (b) | Ratio a/b |
|---|---|---|---|
| Edge density on facades | 10.7067 | 1050.1234 | 0.0102 |
| Edge density on Sky | 9.7539 | 622.3140 | 0.0157 |
| Percent of landscape on facades | 0.0030 | 0.4060 | 0.0075 |
| Percent of landscape on Sky | 0.0016 | 0.0835 | 0.0188 |
| Green Cover on facades | $1.2533 \times 10^{-5}$ | $3.5 \times 10^{-6}$ | 3.5792 |
| Green Cover on Sky view | $7.1352 \times 10^{-6}$ | $3.5016 \times 10^{-6}$ | 2.0377 |
| Number of pixel Landscape metrics | 3,553,123 | | |
| Degree of freedom landscape metrics | 3,552,772 | | |
| N. of pixel Street View metrics | 58,218 | | |
| Degree of freedom Street View metrics | 57,867 | | |
| Degree of freedom Blocks | 351 | | |

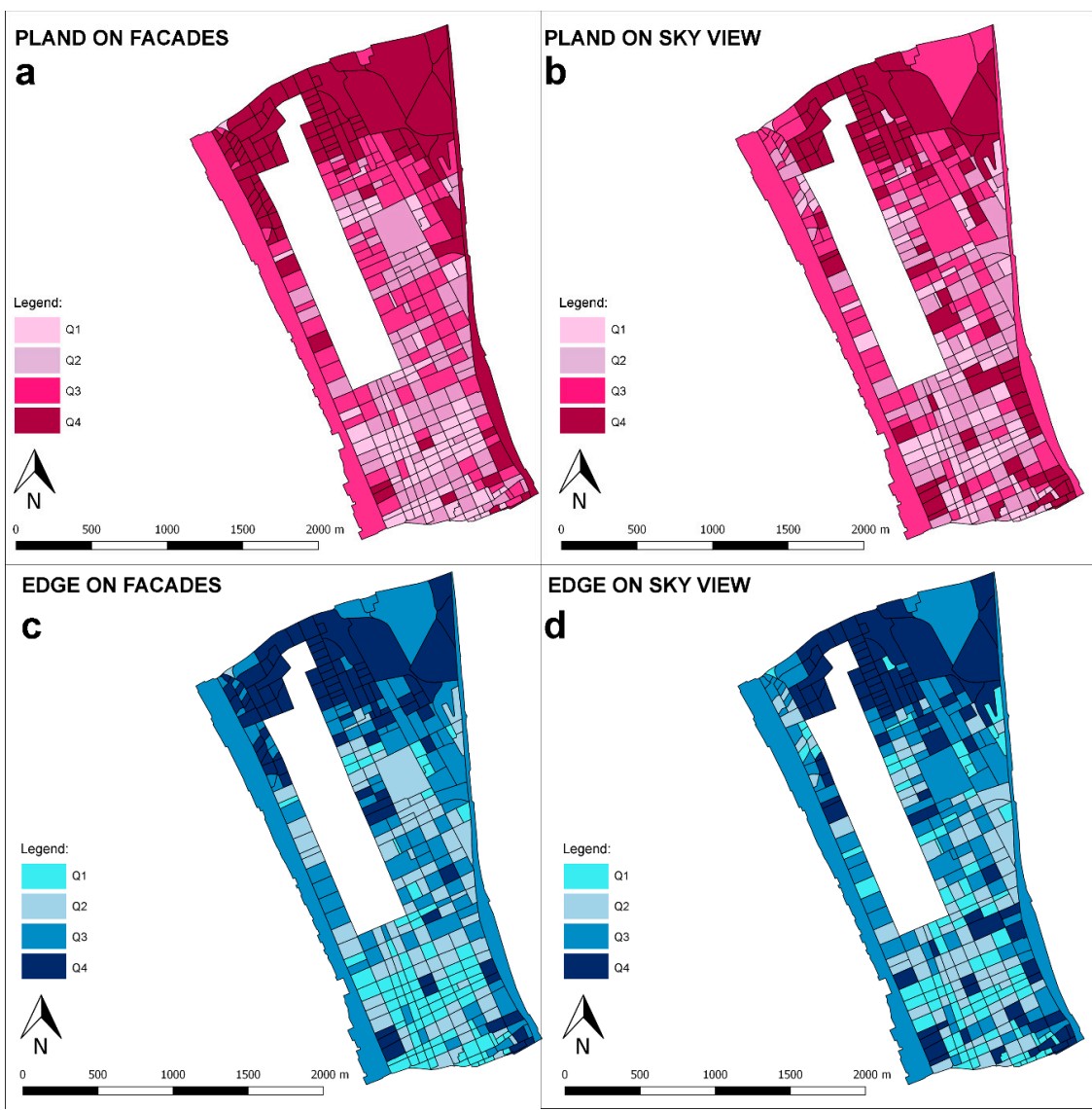

**Figure 9.** Maps of remote sensing metrics: (**a**) percent of landscape on facades; (**b**) percent of landscape on Sky; (**c**) edge density on facades; (**d**) edge density on Sky.

### 3.3. Results of Spatial Clustering with REDCAP Method

As it is necessary to create homogeneous areas by clustering city blocks, Pakzad and Salari [37] explained the ecological and visual characteristics of the city on a larger scale than that of the city block. Therefore, we applied REDCAP as the regionalization tool, which enabled the city to be divided into homogeneous and spatially close clusters based on the same green cover characteristics. The metrics used were: PLAND over block height (PLANDOvB) and PLAND below block height (PLANDBelB), ED over block height (EdgeOvB) and ED below block height (EdgeBelB), and street view on facades (GCF) and street view on sky view (GCS).

Figure 10 shows the correlation between metrics used.

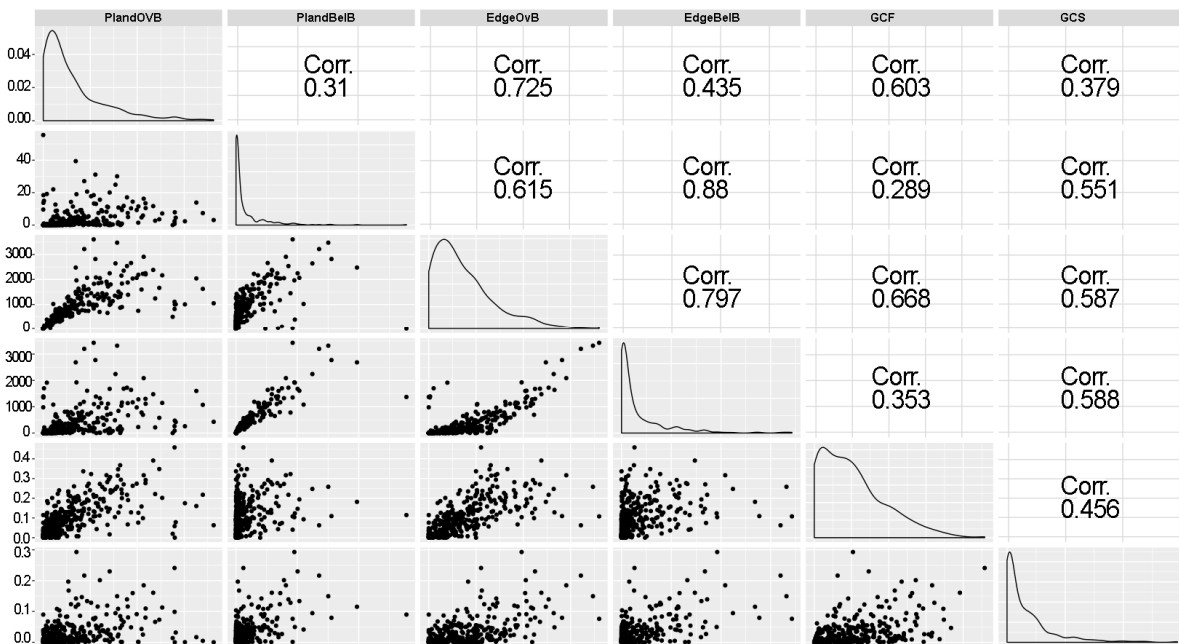

**Figure 10.** Correlation matrix. Main diagonal graph are the frequency distribution of variables. Cells above the main diagonal show the correlation coefficients. Graph above the main diagonal is the scatter plot of the variables.

The regionalization approach is sensitive to correlated attributes during the calculation of homogeneity between regions, and many of the attributes are derived from the same data, leading to a correlation between attributes. Thus, it was necessary to apply principal component analysis (PCA) to the variable's metrics, and unique information could then be retained while excluding correlated information between variables [38–40]. We chose the first three dimensions based on the 95% threshold criterion [41]. Table 3 shows the loading variable of the first three PCs. The first component loads positively on ED metrics, the second PC loads negatively on the metrics below the level of the isolates (GCF, PLANDBelB) and positively with PLANDOvB, and the third PC has a high GCS load.

**Table 3.** Principal component analysis (PCA) dimensions.

|  | PC1 | PC2 | PC3 |
|---|---|---|---|
| Green Cover on Facades | 0.3559 | −0.5382 | 0.2702 |
| Green Cover on Sky | 0.3834 | 0.1345 | 0.8119 |
| Percent of landscape below block height | 0.3686 | −0.4999 | −0.3795 |
| Percent of landscape over block height | 0.4011 | 0.5186 | −0.1583 |
| Edge density below block height | 0.4789 | −0.1277 | −0.1980 |
| Edge denisty over block height | 0.4475 | 0.3964 | −0.2440 |
| % of Variance | 63.0674 | 23.8025 | 9.1246 |

We therefore used the values of PC1, PC2, and PC3 calculated for each of the 351 isolates as input data to identify clusters of contiguous and homogeneous blocks through the REDCAP method. Since the REDCAP method requires to specify a priori the number of clusters to be created, it was necessary to find the optimal number of clusters. We selected the elbow method to determine the optimal number of clusters, as this method optimizes the variance within clusters [42]. This method looks at the variance within the clusters as a function of the number of clusters: One should choose a number of clusters so that adding another cluster does not give much better modeling of the data. More precisely, if one plots the variance within the clusters against the number of clusters, the first clusters will add much information (explain a lot of variance), but at some point the marginal gain will

drop, giving an angle in the graph. The number of clusters is chosen at this point, hence the "elbow criterion". The elbow diagram in Figure 10 shows that when there are more than 10 clusters, there is no significant decrease in the variance within the clusters. So, the optimal number of clusters was 10.

Figures 11b and 12a respectively, show the frequency distribution of ecological metrics within the clusters of homogeneous blocks identified through the REDCAP procedure. Cluster values are shown in Table 4. The lowest value for all metrics is visible in cluster 2, which includes only built city blocks. Furthermore, clusters 1 and 10 are urban blocks with a low presence of vegetation; although they are similar to each other, they belong to two different clusters because they are not contiguous. This characteristic is typical of the geographically constrained clustering methodology REDCAP, which can identify clusters of similar blocks in the metric values as long as they are distant in the geographical space. The maximum values, which are representative of a greater presence and distribution of vegetation, are identified in clusters 9, 7, and 3. These clusters are within the northern area, where there is a greater percentage of greenery and where hedges and rows of large trees are higher than the facades of buildings.

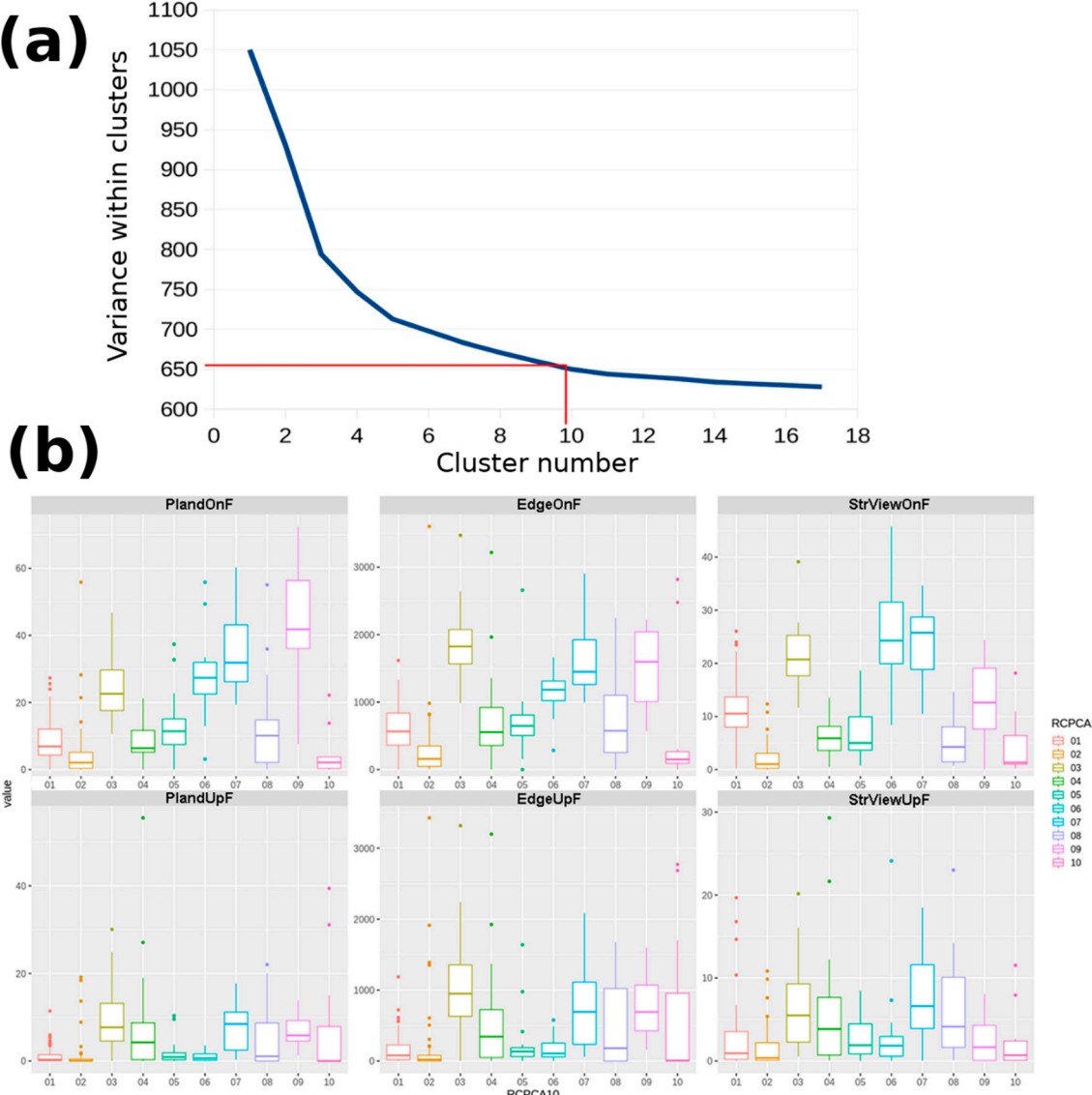

**Figure 11.** Spatial clustering results. (**a**) Elbow method and choice of the optimal number of clusters. (**b**) Frequency distribution boxplot of the green metrics of the blocks within each cluster.

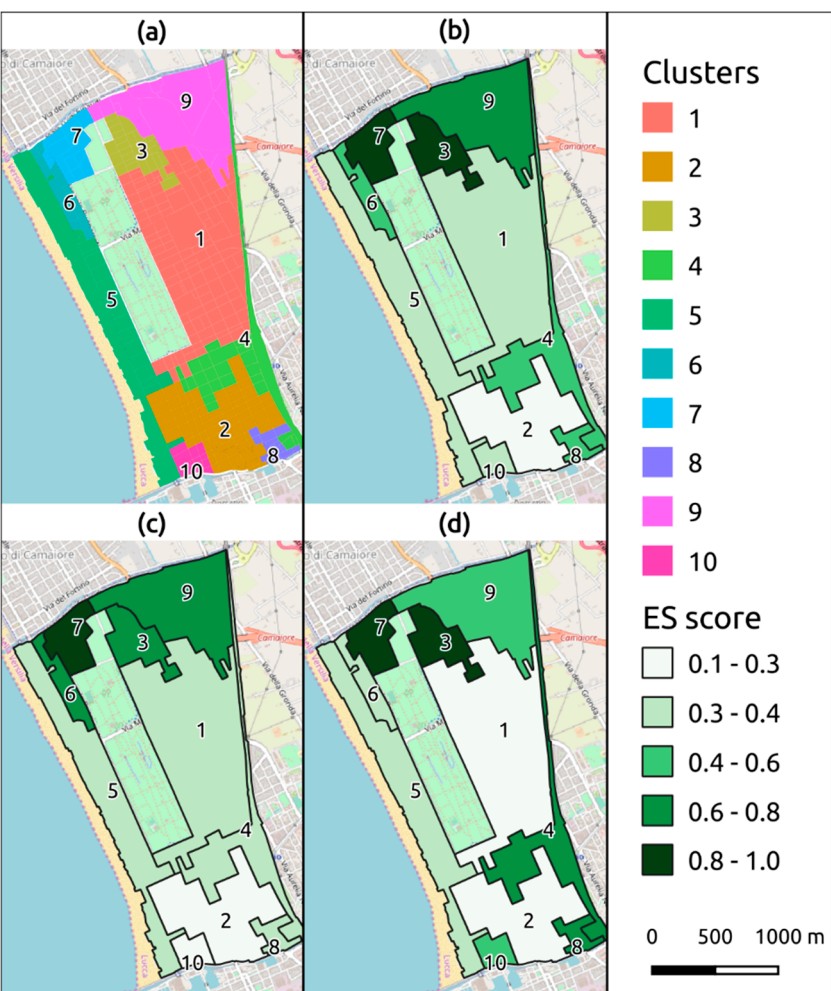

**Figure 12.** (**a**) Cluster map, (**b**) Ecosystem Services total score map, (**c**) Ecosystem Services score below block heights, and (**d**) Ecosystem Services score above the block height.

To summarize the results obtained from the 6 metrics, we proposed the use of three Ecosystem Services (ES) score indices: ES score below block heights ($ES^k_{below}$), ES score above block heights ($ES^k_{above}$), and ES total score ($ES^k_{tot}$), which are formulated as follows:

$$ES^k_{below} = \frac{\sum \frac{PlandBelB_k}{max_k(PlandBelB_k)} + \frac{EdgeBelB_k}{max_k(EdgeBelB_k)} + \frac{GCF_k}{max_k(GCF_k)}}{3} \tag{8}$$

$$ES^k_{above} = \frac{\sum \frac{PlandOverB_k}{max_k(PlandOverB_k)} + \frac{EdgeOverB_k}{max_k(EdgeOverB_k)} + \frac{GCS_k}{max_k(GCS_k)}}{3} \tag{9}$$

$$ES^k_{tot} = \frac{ES^k_{below} + ES^k_{above}}{2} \tag{10}$$

Figure 12b–d shows maps of the three ES indices.

**Table 4.** Mean, median, first, and tirth quartiles of metrics of the blocks within each cluster. PLANDBelB and PLANDOvB are percent of green detected by remote sensing respectively, below and over the height of block. EdgeBelB and EdgeOvB are the edge density of green detected by remote sensing respectively, below and over the height of block. GCF and GCS are the percent of green detected by segmentation of GSV images respectively, on facades and on sky view.

|  |  | Clusters | | | | | | | | | |
|---|---|---|---|---|---|---|---|---|---|---|---|
|  |  | **1** | **2** | **3** | **4** | **5** | **6** | **7** | **8** | **9** | **10** |
| PLANDBelB | Mean | 8.3 | 3.9 | 24.4 | 8.4 | 13.0 | 27.4 | 34.5 | 13.4 | 45.6 | 4.6 |
|  | Q1 | 4.3 | 0.4 | 17.6 | 5.2 | 7.4 | 22.5 | 26.2 | 2.1 | 36.1 | 0.4 |
|  | Median | 6.9 | 2.1 | 22.6 | 6.4 | 11.4 | 27.4 | 31.8 | 10.1 | 41.8 | 2.1 |
|  | Q3 | 12.1 | 5.1 | 29.7 | 11.7 | 15.1 | 32.0 | 43.2 | 14.8 | 56.4 | 3.8 |
| PLANDOvB | Mean | 1.1 | 1.2 | 9.5 | 7.3 | 1.9 | 1.0 | 8.1 | 5.9 | 6.6 | 7.9 |
|  | Q1 | 0.1 | 0.0 | 4.6 | 0.3 | 0.3 | 0.2 | 2.5 | 0.0 | 4.5 | 0.0 |
|  | Median | 0.3 | 0.1 | 7.7 | 4.3 | 0.9 | 0.6 | 8.5 | 1.1 | 5.9 | 0.0 |
|  | Q3 | 1.4 | 0.5 | 13.2 | 8.7 | 1.9 | 1.6 | 11.2 | 8.7 | 9.2 | 7.9 |
| EdgeBelB | Mean | 608.8 | 253.1 | 1884.1 | 768.8 | 706.7 | 1146.3 | 1608.3 | 769.5 | 1493.7 | 593.9 |
|  | Q1 | 362.5 | 48.2 | 1566.5 | 355.9 | 505.7 | 1020.4 | 1262.2 | 252.0 | 1006.3 | 93.3 |
|  | Median | 565.6 | 157.8 | 1824.9 | 554.0 | 646.5 | 1183.5 | 1449.2 | 574.8 | 1596.8 | 150.5 |
|  | Q3 | 838.2 | 349.0 | 2073.6 | 919.6 | 807.7 | 1311.6 | 1924.1 | 1099.8 | 2041.3 | 266.3 |
| EdgeOvB | Mean | 152.0 | 150.2 | 1053.5 | 540.3 | 238.0 | 165.3 | 720.9 | 498.2 | 769.2 | 672.8 |
|  | Q1 | 18.0 | 0.0 | 631.2 | 50.6 | 64.1 | 59.0 | 236.0 | 0.0 | 426.9 | 0.0 |
|  | Median | 80.0 | 21.5 | 950.4 | 344.2 | 134.4 | 105.7 | 693.9 | 180.5 | 692.3 | 6.3 |
|  | Q3 | 226.9 | 80.4 | 1354.7 | 724.0 | 187.5 | 252.8 | 1111.5 | 1021.9 | 1071.1 | 956.3 |
| GCF | Mean | 11.1 | 2.0 | 21.5 | 6.0 | 7.3 | 26.0 | 23.6 | 5.1 | 12.9 | 4.6 |
|  | Q1 | 8.0 | 0.2 | 17.7 | 3.6 | 3.7 | 19.9 | 18.8 | 1.5 | 7.6 | 1.0 |
|  | Median | 10.5 | 1.0 | 20.7 | 5.9 | 5.0 | 24.3 | 25.8 | 4.3 | 12.6 | 1.3 |
|  | Q3 | 13.7 | 3.1 | 25.3 | 8.1 | 10.0 | 31.5 | 28.7 | 8.0 | 19.1 | 6.4 |
| GCS | Mean | 2.2 | 1.4 | 6.8 | 5.6 | 2.7 | 3.1 | 8.1 | 6.7 | 2.6 | 2.4 |
|  | Q1 | 0.2 | 0.1 | 2.3 | 0.7 | 0.9 | 0.6 | 3.9 | 1.6 | 0.1 | 0.1 |
|  | Median | 0.9 | 0.4 | 5.5 | 3.9 | 1.9 | 1.9 | 6.6 | 4.1 | 1.7 | 0.7 |
|  | Q3 | 3.6 | 2.2 | 9.3 | 7.7 | 4.5 | 3.0 | 11.6 | 10.1 | 4.3 | 2.4 |

The figures show that areas with the lowest scores for the three ES indices are clusters 1, 2, and 5. The scores are particularly low in cluster 5 (see Figure 13a), which includes the promenade and is the most important tourist area of the city. Clusters 3, 6, 7, and 9 have the highest scores for all three indices and are representative of the so-called "garden city" which was recently developed. The high values for the above-block height scores occur in relation to plants preserved in the pine forest within the urban area (Figure 13b). Clusters 1, 2, 4, 8, and 10 are all characterized by a high building density, and slightly different scores are obtained. For example, there are rows of small plants along the sidewalks (Figure 13c) or trees in the middle of roads (Figure 13d) in clusters 4 and 8, whereas clusters 1, 2, and 10 have almost no public greenery, despite having a similar urban fabric.

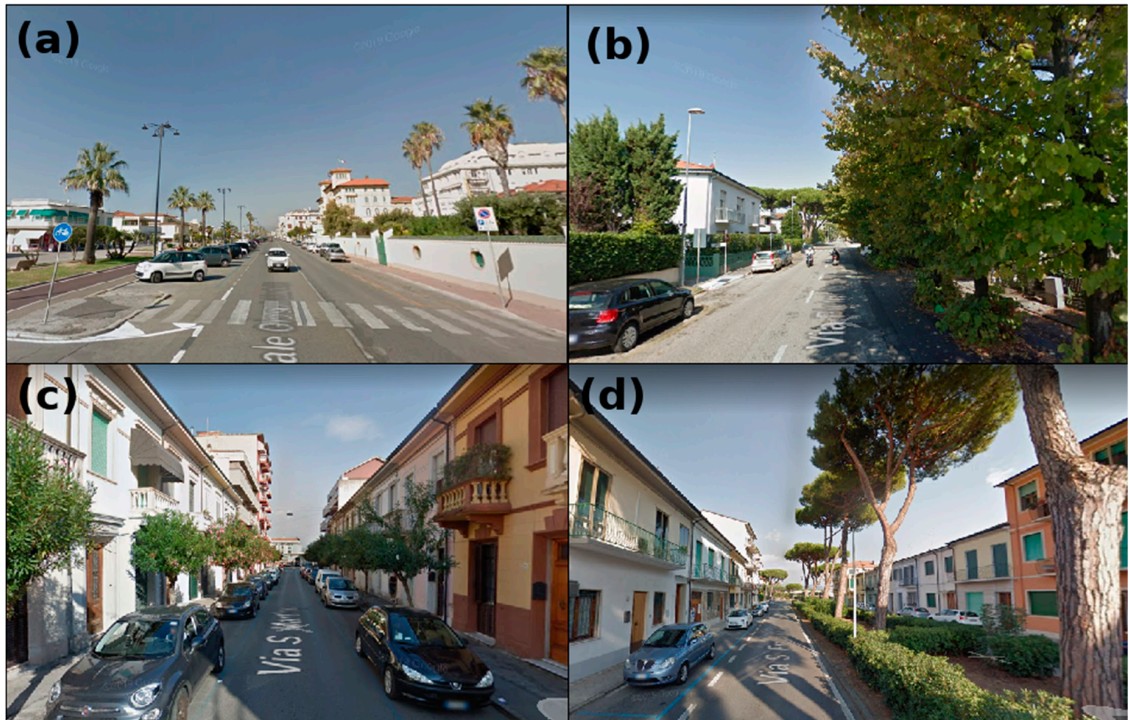

**Figure 13.** Typical representative public greenery: (**a**) promenade, (**b**) pine forest within the urban area, (**c**) are rows of small plants along the sidewalks, (**d**) trees in the middle of roads.

## 4. Discussion

This paper proposes a set of ecological metrics for urban greenery that can be used to evaluate urban green ecosystem services. The metrics were derived from two complementary surveys: a traditional remote sensing survey using multispectral images and LiDAR data, and a proximate sensing survey using images available on the GSV database.

With respect to the classification of vegetated areas, the results of our study confirm the efficacy of the combined use of NDVI and LiDAR data for remotely-sensed images [43,44] and of semantic segmentation using deep learning for GSV images [45,46].

The methodology is unique, as it identifies metrics (based on existing literature) of urban green ecosystem services using both data sources. The ecosystem services of urban forests were derived from greenery shading facades of buildings and from the covering of roads, roofs, and courtyards by the foliage of trees. The metrics were then divided into urban greenery below the height of buildings and that above the height of buildings. The correlation matrix in Figure 10 shows that the proximate sensing metrics agree with those derived from remote sensing data. The partial correlation (about 0.6) between GCF/GCS and the PLAND/ED indices shows that both reliefs can map the presence of vegetation with good coherence, but they reveal different characteristics. Proximate sensing data more efficiently highlights the shading of facades and streets, but the data are limited to public spaces that are accessible via a vehicle. In contrast, the data from remote sensing are more efficient for highlighting roofing and the shading of private courtyards.

The urban green survey using proximate sensing images at street level can be conducted at a low cost and can thus be used to monitor the health status of urban green spaces. However, further research is required to define methodologies (based on the use of commercial spherical cameras integrated with medium-scale satellite images) that can be used by city administrations (Landsat and Sentinel) [45,46] to conduct low-cost, short-interval monitoring.

In accordance with previous studies [47–50], we report a set of metrics for city blocks. The research also made it possible to create a methodology for verifying the ecological homogeneity of urban blocks which can therefore be considered efficient tessellation units for managing urban green areas.

Indeed Sheppard et al. [51] argue that the city block scale is an often neglected but promising level for community engagement and co-creation of climate change responses.

However, an innovative element of this study is the application of regionalization techniques through constrained clustering that are used to identify homogeneous areas, with the aim of determining the horizontal and vertical characteristics of urban greenery. This method will enable urban planners to identify areas within a city that can be considered greener than others, and it can also be used as a monitoring tool when conducting a differentiated analysis of gain or loss with respect to urban street greenery. In addition, it could be used in the planning stage of an urban greening program to assist urban planners in selecting appropriate locations, sizes, and types of greenery that provide the maximum affect. Furthermore, it could be employed to check the visual impact of urban forest management practices and document the visibility of urban greenery in cities. The work also showed that the block is a sufficiently homogeneous and therefore efficient geographical unit for the classification of urban green. A finer geographical unit could have the advantage of a greater homogeneity of the metrics, but it would make it more difficult to transfer the results of the research to guide management and improvement interventions of the green ecosystem services in the city.

The block clusters obtained with the REDCAP method in fact meet the requirements set by Barron et al. [52] (p. 21) to define the "neighborhood scale" as a set of cohesive blocks. They highlight that "The neighborhood scale captures human green space experiences at shorter distances, allowing for consideration of accessibility, sightlines, aesthetics, vegetation layering, and quality of greenspace design. The experiential neighborhood scale is small enough to conceptually include the impact of individual trees, an important component of urban greens paces". Therefore, the authors in their work propose interventions that provide strategic green space enhancement at the neighborhood and block scale.

The high diversification and complementarity of the metrics proposed at the city block level enable a better understanding of the role that trees and vegetation play in urban dynamics and human health. Although some studies have shown a link between human health benefits and the presence of urban greenery [53–55], these studies have used small samples and a limited geographical scope for a single route. The ability to measure a complete set of ecological green urban metrics would enable researchers to determine whether the health benefits of urban trees are pervasive, whether they exist in specific contexts with respect to biogeographical conditions, or whether they can be maximized (for example with respect to local policies, management practices, and socioeconomic indicators). This method enables urban tree cover and its relationship with local conditions or social factors to be analyzed on a much finer scale than previously possible. The relationships between urban trees and physical and social components of cities that were previously opaque, such as how income level and social status relate to tree presence and neighborhood aesthetics, can now be investigated in depth.

## 5. Conclusions

Our study proposes a set of ecological green urban metrics based on the integration of remote sensing and proximate sensing data. Metrics derived from proximate sensing were calculated by classifying the urban forest using GSV database images. Green areas were classified by semantic segmentation using pretrained deep neural networks. The results demonstrate the high efficiency of the method, which has an accuracy of 83%.

Metrics from remote sensing were derived from overlaying multi-spectral images and LiDAR data. In accordance with previous studies, two classes of metrics were calculated: greenery at a lower or higher elevation than building facades, respectively.

In the last phase of the work, the metrics were related to city blocks, and homogeneous areas were identified relative to the characteristics of urban green using a spatially constrained clustering methodology.

The proposed methodology enables the creation of a geographic information system that can be used by public administrators and urban green designers to maintain and create new public forests in cities.

Furthermore, research can be further developed in three possible ways: by expanding the survey to other urban elements, in particular buildings and roads, by verifying the relationships between homogeneous clusters that show characteristics of landscape ecology and the visual quality of the city, climatic well-being, or as a pollution reduction strategy, or by employing different and more complex landscape metrics.

**Supplementary Materials:** The following are available online at http://www.mdpi.com/2072-4292/12/2/329/s1. grassProcedure.txt: GRASS procedure. MATLAB_procedure.m: MATLAB procedure. R_procedure.R: Cran R procedure. SegExamp.zip: Sample of segmented Google Street View images. CityBlock.xlsx: descriptive statistics of city blocks metrics.

**Author Contributions:** Conceptualization, I.B., I.C., and E.B.; methodology, I.B. and E.B.; software, I.B. and E.B.; validation, I.B. and C.S.; writing—original draft preparation, I.B.; writing—review and editing, E.B. and I.B.; visualization, E.B. and I.C.; supervision, C.S.; funding acquisition, C.S. The manuscript was written by I.B. and improved by the contributions of all the co-authors. All authors have read and agreed to the published version of the manuscript.

**Funding:** The authors acknowledge financial support from the "Unione dei comuni Circondario dell'Empolese Valdelsa".

**Conflicts of Interest:** The authors declare no conflicts of interest.

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
