# Peer review of "Integrating Remote Sensing and Street View Images to Quantify Urban Forest Ecosystem Services"

_remotesensing, doi:10.3390/rs12020329_

Round 1

Reviewer 1 Report

The topic is little with satellite data though the special issue is with integration of satellite remote sensing and spatial information. Therefore, I would recommend this for submission to other journals, such as ISPRS International Journal of Geo-Information. For the future submission, I would like the authors to address my comments as below.

1. Section 3.2 and Figure 8 - Because homogeneity within each city block are not validated, please consider to present standard deviation/variable or a box plot for each city block.
2. L191 - Open Street Map --> OpenStreetMap
3. Figure 5 - Please add descriptions for (a) and (b) in the caption.
4. Figure 6 - Please add descriptions for a-c in the caption. The axis labels are blurred, or too small to read.
5. Figure 9 - Please enlarge the numbers.
6. Figure 9 - What does the curves in the diagonal elements indicates? Please add descriptions about the charts in the caption.
7. Table 2 - Please add a row for contribution (%) of the components and also discuss the possible improvements and benefits by finer gratuity.

Author Response

Review 1.

We thank first of all the reviewer for the suggestions that have allowed to improve the work. Above all, the observation relating to the evaluation of the homogeneity of the blocks inspired a new elaboration that added important elements to the methodology. We are sorry but, due to our poor English, we have not been able to fully understand the comment n.7. The following specific answers to the questions.

Q.

The topic is little with satellite data though the special issue is with integration of satellite remote sensing and spatial information. Therefore, I would recommend this for submission to other journals, such as ISPRS International Journal of Geo-Information. For the future submission, I would like the authors to address my comments as below.

A.

We thank the referee for the suggestion. We know the International Journal of Geo-Information and believe that it is certainly a prestigious magazine (we currently have a paper under review on it). Moreover, the integration of remote sensing data from satellite and proximate sensing represents a frontier argument that overlaps the themes of remote sensing and spatial information. Remote Sensing has recently published works based on the synergy between information deriving from the segmentation of Google Street View images and remote sensing data (see Cao, Rui, et al. "Integrating Aerial and Street View Images for Urban Land Use Classification." Remote Sensing 10.10 2018 and Stubbings, Philip, et al. "A Hierarchical Urban Forest Index Using Street-Level Imagery and Deep Learning." Remote Sensing 11.12 2019). The work we present represents the continuation of this line of research. Furthermore, our work derives from the development of the research presented at the RSCy2019 conference from which the special issue derives. For these reasons, if possible we would prefer to publish on the special issue.

Q.

Section 3.2 and Figure 8 - Because homogeneity within each city block are not validated, please consider to present standard deviation/variable or a box plot for each city block.

A.

We agree with the referee's observation. In fact, the verification of the homogeneity of the metrics within the blocks allows to verify the efficiency of the blocks as a tesselation element of the study area. For this reason, we have implemented a specific analysis of variance within and between blocks. Statistics were calculated within and among the blocks in order to verify their homogeneity. The following text has been added: “In order to evaluate the homogeneity of the metrics within each block were calculated by the standard deviation statistics within and between the blocks. In fact, the verification of the homogeneity of the metrics within the blocks allows to verify the efficiency of the blocks as a tesselation element of the study area. The results shown in table 2 show that the blocks are an efficient tessellation of the urban space. The descriptive statistics of the 351 isolates are available as supplementary material.” Table 2 has been added. The mean and variance for each of the 351 isolates are available as supplementary material in an excel file. The following sentences has been added in Discussion section.

“The research also made it possible to create a methodology for verifying the ecological homogeneity of urban blocks which can therefore be considered efficient tesselation units for managing urban green areas. Indeed Sheppard et al. [47] argue that the city block scale is an often neglected but promising level for community engagement and co-creation of climate change responses.” And “The block clusters obtained with the REDCAP method in fact meet the requirements set by Barron et al. [52] to define the “neighborhood scale” as a set of cohesive blocks. They highlight that “The neighborhood scale captures human green space experiences at shorter distances, allowing for consideration of accessibility, sightlines, aesthetics, vegetation layering, and quality of greens pace design. The experiential neighborhood scale is small enough to conceptually include the impact of individual trees, an important component of urban greens paces” Therefore the authors in their work propose interventions that provide strategic green space enhancement at the neighborhood and block scale.”

Q.

L191 - Open Street Map --> OpenStreetMap

A.

Done.

Q.

Figure 5 - Please add descriptions for (a) and (b) in the caption.

A.

Done.

Q.

Figure 6 - Please add descriptions for a-c in the caption. The axis labels are blurred, or too small to read.

A.

Done.

Q.

Figure 9 - Please enlarge the numbers.

A.

Q.

Figure 9 - What does the curves in the diagonal elements indicates? Please add descriptions about the charts in the caption.

A.

Done.

Q.

Table 2 - Please add a row for contribution (%) of the components and also discuss the possible improvements and benefits by finer gratuity.

A.

% of contribution: done. We are sorry but we cannot understand what "and also discuss the possible improvements and benefits by finer gratuity" means for table 2. If instead it is a suggestion that refers to the Discussion paragraph we have inserted future improvements in the Conclusions paragraph.

Reviewer 2 Report

This paper is about integrating remote sensing and urban aerial (satellite) multispectral and LiDIAR image in terrain (objects) classification application, with properly validated results but with a not clear foundation and justification.

I suggest the following either MAJOR concerns or minor comments:

MAJOR CONCERNS:

1) I do not see clear the interest of combining city urban (urban forest, street/buildings) and green open space (urban greenery, gardens) imaging data altogether. An effort in better justifying the interest of the present approach is needed.

2) In addition to confusion matrices (c.f. Table 1), I suggest to also include Receiver Operation Curves curve analsys and Area Under the ROC Curve (ACU) computation as a relevant classification index:

https://towardsdatascience.com/understanding-auc-roc-curve-68b2303cc9c5

https://www.sciencedirect.com/science/article/pii/S016786550500303X

Please note that despite ROC and AUC are defined in principle for binary classification problems, it is also valid for multiple-classification (multiple output clases).

3) I would like to see a number of representative output segmented/classified images combining the various type of either urban or green objects (urban forest, urban buildings, urban green spaces, etc), at least as an Appendix of supplementary data for this mansucript.

Minnor comments:

1) Figure and table captions are too short, see for instance Fig.10 and Table 3. Please provide enough detail to all figure and table captions, to avoid the reader to seek inside manuscript text in order to understand a figure.

2) Further details of Materials is needed. Please provide.

3) Please provide details about train/test sets and proper validation of the simulations and results carried out.

4) Following paragrahp need to be better expplained and rewritten:

"Figures 10b and 11a respectively show the frequency distribution of ecological  metrics within homogeneous regions and the cluster map. The values of the clusters are shown in Table 3. The lowest value for all metrics is seen in cluster 2,  which exclusively comprises built-up city blocks. Furthermore, clusters 1 and 10 are urban blocks with a low vegetation presence; although they are similar to each other, they belong to two different clusters because they are not contiguous.  This characteristic is typical of the spatialization method used, which identifies similar but distant groups"

What is  each "cluster"? Why are there 10 clusters? What is the "spatialization method used"?

Author Response

We thank first of all the reviewer for the suggestions that have allowed to improve the work. Above all major revision n.2 added important elements to the evaluation of the results. The following specific answers to the questions.

 Q.

1) I do not see clear the interest of combining city urban (urban forest, street/buildings) and green open space (urban greenery, gardens) imaging data altogether. An effort in better justifying the interest of the present approach is needed.

A.

In the introduction we added the following paragraph which we hope will clarify the referee's doubts.

"Earth observation data such as multispectral satellite images have long been used to classify green open spaces in cities. Despite the large increase in spatial resolution, urban green classification from aerial images is still considered a difficult task, since only the upper part of plants can be captured from the nadir's point of view. For this reason, high resolution satellite images are useful for classifying urban green with a wide spatial extension, such as urban forests, green parks and gardens, but they are not efficient for urban green scattered or in rows. In fact, the lack of detail at ground level makes it difficult to detect the characteristics deriving from the shape of the foliage of the plants. Therefore the urban green classification at street level is still based on a labor-intensive territorial survey, which is inefficient and expensive. Fortunately, the growing accessibility to different geo-tagged data sources allows to merge remote sensing images with data of different modalities and observations. For example, Google Street View (GSV) offers users street-level panoramic images captured in thousands of cities around the world, which allows you to observe street scenes in big cities and thus provides instant detection capabilities and detail at the level of the soil that lack of aerial imagery."

Q.

2) In addition to confusion matrices (c.f. Table 1), I suggest to also include Receiver Operation Curves curve analsys and Area Under the ROC Curve (ACU) computation as a relevant classification index:

https://towardsdatascience.com/understanding-auc-roc-curve-68b2303cc9c5

https://www.sciencedirect.com/science/article/pii/S016786550500303X

Please note that despite ROC and AUC are defined in principle for binary classification problems, it is also valid for multiple-classification (multiple output clases).

A.

The ROC curves and AUC parameters were calculated. We added the following paragraph.

“Figure 6 shows the Receiver Operation Curves (ROC) curves and the values of the area under the ROC curve (AUC) for the different classes calculated by means of a sub-sample of 50 images (due to . The area under the curve for all the four class is very large indicating the high accuracy of the algorithm.

These results are in line with those obtained in other researches that have used DeepLabV3 + in remote sensing [34] or in other fields of study [35]. Few segmentation outputs obtained using DeepLabV3+ are in supplementary materials. It can be seen that the green class is segmented accurately with a sharp classboundary.”

Q.

3) I would like to see a number of representative output segmented/classified images combining the various type of either urban or green objects (urban forest, urban buildings, urban green spaces, etc), at least as an Appendix of supplementary data for this mansucript.

A.

The required images have been added to the supplementary material. See also the last sentence in the answer to previous comment.

Minnor comments:

Q.

1) Figure and table captions are too short, see for instance Fig.10 and Table 3. Please provide enough detail to all figure and table captions, to avoid the reader to seek inside manuscript text in order to understand a figure.

A.

Done. Please note that Figure 10 is now renamed as figure 11 and table 3 is now renamed as table 4.

Q.

2) Further details of Materials is needed. Please provide.

A.

Further information has been added in the Material and Methods paragraph both in the study area subsection and in the data sources subsection.

Q.

3) Please provide details about train/test sets and proper validation of the simulations and results carried out.

Done. We added the following paragraph in section 2.1.2. Image segmentation

“To validate the network, we extracted 200 random images from those downloaded with Street View API, manually segmented them, and then used them as a validation set to evaluate the performances of the pretrained network. Deeplab v3+ is trained using 60% of the images from the dataset. The rest of the images are split evenly in 20% and 20% for validation and testing respectively.”

Q.

4) Following paragrahp need to be better expplained and rewritten:

"Figures 10b and 11a respectively show the frequency distribution of ecological  metrics within homogeneous regions and the cluster map. The values of the clusters are shown in Table 3. The lowest value for all metrics is seen in cluster 2,  which exclusively comprises built-up city blocks. Furthermore, clusters 1 and 10 are urban blocks with a low vegetation presence; although they are similar to each other, they belong to two different clusters because they are not contiguous.  This characteristic is typical of the spatialization method used, which identifies similar but distant groups"

What is  each "cluster"? Why are there 10 clusters? What is the "spatialization method used"?

A.

We agree with the reviewer. The whole section  was not easily understood and has been extensively rewritten and enlarged as follow, hoping that will clarify the reviewers' doubts.

“We therefore used the values of PC1, PC2 and PC3 calculated for each of the 351 isolates as input data to identify clusters of contiguous and homogeneous blocks through the REDCAP method. Since the REDCAP method requires to specify a priori the number of culsters to be created, it was necessary to find the optimal number of clusters. We selected the elbow method to determine the optimal number of clusters, as this method optimizes the variance within clusters [40]. This method looks at the variance within the clusters as a function of the number of clusters: One should choose a number of clusters so that adding another cluster doesn't give much better modeling of the data. More precisely, if one plots the variance within the clusters against the number of clusters, the first clusters will add much information (explain a lot of variance), but at some point the marginal gain will drop, giving an angle in the graph. The number of clusters is chosen at this point, hence the "elbow criterion". The elbow diagram in Figure 10a shows that when there are more than 10 clusters, there is no significant decrease in the variance within the clusters. So the optimal number of clusters was 10.

Figures 11b and 12a respectively show the frequency distribution of ecological metrics within the clusters of homogeneous blocks identified through the REDCAP procedure. Cluster values are shown in Table 3. The lowest value for all metrics is visible in Cluster 2, which includes only built city blocks. Furthermore, clusters 1 and 10 are urban blocks with a low presence of vegetation; although they are similar to each other, they belong to two different clusters because they are not contiguous. This characteristic is typical of the geographically constrained clustering methodology REDCAP, which can identify clusters of similar blocks in the metric values as long as they are distant in the geographical space.”.

Round 2

Reviewer 1 Report

Thank you for the responses. I confirmed my comments have been addressed in the revision. Regarding "and also discuss the possible improvements and benefits by finer gratuity", this is my mistake -- the comments should be under "Section 3.2 and Figure 8 - Because homogeneity within each city block are not validated, please consider to present standard deviation/variable or a box plot for each city block." I am sorry for the misleading comment.

Reviewer 2 Report

I am glad to say that authors have responded to my queries in a proper fashion modifyimg manuscripr accordingly and I believe manuscript has consequently been improved significatevely with the inclusion of results validation.